# Effects of Mind–Body Exercise on Brain Structure and Function: A Systematic Review on MRI Studies

**DOI:** 10.3390/brainsci11020205

**Published:** 2021-02-07

**Authors:** Xiaoyou Zhang, Boyi Zong, Wenrui Zhao, Lin Li

**Affiliations:** 1Key Laboratory of Adolescent Health Assessment and Exercise Intervention of Ministry of Education, East China Normal University, Shanghai 200241, China; 52191000016@stu.ecnu.edu.cn (X.Z.); 52201000012@stu.ecnu.edu.cn (B.Z.); 52181000016@stu.ecnu.edu.cn (W.Z.); 2College of Physical Education and Health, East China Normal University, Shanghai 200241, China

**Keywords:** mind–body exercise, MRI, brain structure, brain function, systematic review

## Abstract

Mind–body exercise has been proposed to confer both physical and mental health benefits. However, there is no clear consensus on the neural mechanisms underlying the improvements in health. Herein, we conducted a systematic review to reveal which brain region or network is regulated by mind–body exercise. PubMed, Web of Science, PsycINFO, SPORTDiscus, and China National Knowledge Infrastructure databases were systematically searched to identify cross-sectional and intervention studies using magnetic resonance imaging (MRI) to explore the effect of mind–body exercise on brain structure and function, from their inception to June 2020. The risk of bias for cross-sectional studies was assessed using the Joanna Briggs Institute (JBI) checklist, whereas that of interventional studies was analyzed using the Physiotherapy Evidence Database (PEDro) scale. A total of 15 studies met the inclusion criteria. Our analysis revealed that mind–body exercise modulated brain structure, brain neural activity, and functional connectivity, mainly in the prefrontal cortex, hippocampus/medial temporal lobe, lateral temporal lobe, insula, and the cingulate cortex, as well as the cognitive control and default mode networks, which might underlie the beneficial effects of such exercises on health. However, due to the heterogeneity of included studies, more randomized controlled trials with rigorous designs, similar measured outcomes, and whole-brain analyses are warranted.

## 1. Introduction

Mind–body exercise is a form of multicomponent exercise that combines movement sequences, breathing control, and attention regulation, which is different from traditional physical exercise [1]. It is also referred to as movement-based contemplative practice [2] or mindful movement [3], which emphasizes moving mindfully, commonly including Tai Chi Chuan (TCC), Qigong, and yoga. TCC is a form of mind–body exercise incorporating physical, cognitive, social, and meditative components [4]. Qigong involves a set of relatively slow exercises through coordinated physical movements, breathing, and meditative state to cultivate one’s internal energy called “Qi” to achieve body healing, and Baduanjin (BDJ) is one of the most common forms of Qigong [5]. Yoga is an ancient mind–body exercise which focuses on the present moment, consisting of physical postures (asanas), control of breath (pranayama), and the use of meditation (dyana), and the most common form is Hatha yoga [6]. Compared with aerobic or resistance exercise, mind–body exercises are relatively low in intensity and slow in pace, particularly suitable for the elderly and groups with chronic diseases [7].

In recent years, increasing research evidence has shown that mind–body exercise could improve and promote physical health [8,9,10] as well as benefit mental health, including improving general cognition, executive function, learning, memory, and verbal fluency [11,12,13]. Moreover, it aids in relieving stress [14,15], anxiety, depression, and other negative emotions [16,17] as well as enhancing the subjective well-being of an individual [6,18].

Numerous studies have reported promising results that support the effects of mind–body exercise on health benefits. However, the mechanisms underlying these improvements remain largely unknown. The physical and mental health improvements on the behavior of an individual are often accompanied by changes in the brain structure or function of specific regions or networks [19]. Therefore, understanding the effects of mind–body exercise on the brain plasticity will significantly help to formulate more scientific interventions and, importantly, improve the behavior and brain health level of healthy or clinical populations. The effects of mind–body exercise on brain plasticity have often been examined by magnetic resonance imaging (MRI). MRI is a frequently used non-invasive neuroimaging technique with remarkable spatial resolution that allows for investigation of changes in cortical as well as subcortical brain regions, including two major modalities: structural MRI (sMRI) and functional MRI (fMRI) [20]. sMRI provides measures of cerebral anatomy in vivo [21], and fMRI detects brain activity and network connectivity based on blood oxygenation level-dependent (BOLD) signals [22].

A few studies have assessed structural or functional brain changes regulated by mind–body exercise using MRI. However, the available studies have several limitations such as small sample size and diverse research designs and outcome measures, resulting in low statistical power and challenges in obtaining consistent changes in the brain regions or networks. To address these limitations, evidence-based medicine suggests that systematic reviews or meta-analyses should be used to combine the findings of multiple related primary studies to make them more persuasive [23].

To the best of our knowledge, there is no systematic review or meta-analysis study that has integrated findings focused on different types of mind–body exercise as a whole. Previous systematic reviews on TCC [24] and yoga [19,25], were based on a small number of studies and did not focus on specific brain regions or networks affected by mind–body exercise. In the present study, we conducted a systematic review of MRI-based studies investigating the relationship between mind–body exercise and brain structure and function to elucidate the possible neural mechanisms underlying the health benefits of mind–body exercises. Our findings can provide a theoretical basis for mind–body exercises in promoting the healthy development of the body, mind, and the brain.

## 2. Methods

### 2.1. Literature Search and Study Selection

We performed a systematic electronic literature search in PubMed, Web of Science, PsycINFO, SPORTDiscus, and China National Knowledge Infrastructure (CNKI) databases from their inception to June 2020 to identify relevant studies. The databases were searched for articles published in either English or Chinese language using the following terms: “Tai Chi Chuan”, or “Taiji”, or “Qigong”, or “Baduanjin”, or “Wuqinxi”, or “Yoga”, or “mind-body exercise” in combination with “neuroimaging”, or “fMRI”, or “MRI”. Corresponding Chinese words were used in the Chinese databases. Besides, we explored several other sources, including the bibliography and citation indices of the pre-selected papers and direct searches of the names of frequently cited authors.

All searched records were imported into EndNote X9 (Thomson Reuters), which facilitated the removal of duplicates. Two reviewers (ZX and ZW) independently selected and checked the eligible articles according to the inclusion criteria. Any disagreements were resolved through a discussion with a third reviewer (ZB).

### 2.2. Inclusion and Exclusion Criteria

The inclusion and exclusion criteria of the eligible studies constituted:(i)Participants: study populations consisted of healthy adults or older adults, regardless of sex, racial, and ethnic groups. We excluded studies with subjects who had cognitive impairment or suffering from organic diseases such as diabetes, fibromyalgia, knee osteoarthritis, or tinnitus to avoid interference of the results by these factors.(ii)Type of exercise: the experimental group engaged in mind–body exercises, including Tai Chi Chuan (TCC), Qigong, and yoga. Studies with multimodal interventions comprising mind–body exercises were also included to increase the number of enrolled articles. However, we excluded studies that examined the sole effects of mindfulness or meditation since the aim of this study was to examine the effect of holistic mind–body exercises which involve structured movements, breath control, and attention modulation on brain health.(iii)Study design: to gain an overall understanding of mind–body exercise-related changes in the brain plasticity of healthy adults, there were no restrictions on the study design. Therefore, randomized and non-randomized controlled interventions and within-subjects intervention studies as well as cross-sectional studies comparing experts to novices were all included. With reference to previous studies [25,26], for the intervention studies, the experimental group must have exercised for at least 4 weeks with more than one session per week, and for the cross-sectional studies, the regular exercise duration must have been not less than 3 years to provide sufficient time for changes in the brain structure and function to occur.(iv)Outcomes: the imaging technique was restricted to MRI, including sMRI, task fMRI, and resting-state fMRI. The outcome measures included changes in structure (i.e., gray matter volume, density, and cortical thickness) and task or resting-state (de)activation, functional connectivity for pre- to post-mind–body intervention, or mind–body expert–novice comparison.(v)Literature type: peer-reviewed articles published either in English or Chinese language.

### 2.3. Data Extraction

Two independent researchers (ZX and ZW) performed data extraction from the eligible studies. Any disagreements were resolved via discussions with a third researcher (ZB). For the cross-sectional studies, we extracted data on the sample size, group age, group description, main outcome measures, primary MRI results, and the association with behavior results. At the same time, the extracted data for intervention studies included participants and study design, sample size, group age, intervention frequency and duration, main outcome measures, primary MRI results, and the association with behavior results.

### 2.4. Quality Assessment

We assessed the methodological quality of the included cross-sectional studies using the Joanna Briggs Institute (JBI) checklist for analytical cross-sectional studies [26,27]. The checklist comprised 8 items and possible answers were “yes”, “no”, “unclear”, or “not applicable”. According to relevant studies [28,29], studies were characterized according to the following: (i) low risk of bias if studies reached more than 70% score “yes”; (ii) moderate risk of bias if “yes” scores were between 50% and 69%; and (iii) high risk of bias if “yes” scores were below 49%. For the intervention studies, we used the Physiotherapy Evidence Database (PEDro) scale developed by the Delphi list [30]. The scale consists of 11 items which were scored as either 1 (the answer was “yes”) or 0 (the answers were “no”, “unclear”, or “not applicable”). The studies were classified, using the total rating score (item 1 not scored), as having excellent (9–10), good (6–8), fair (4–5), or poor (<4) quality [13]. In addition, to reduce the risk of bias in the assessment, two researchers (ZX and ZW) scored the quality of the included articles independently. Any conflicting scores between the two researchers were resolved via a discussion with a third researcher (ZB).

## 3. Results

### 3.1. Study Search and Characteristics

Figure 1 shows a flow chart summarizing the study selection process recommended by the PRISMA (preferred reporting items for systematic reviews and meta-analyses) guidelines. Overall, 29 studies focused on evaluating mind–body exercise and brain plasticity using MRI met the inclusion criteria. It is worth noting that in 14 cases, authors published multiple articles on one actually performed study; therefore, we merged the related articles into one study, resulting in 15 actual studies. The characteristics of the included cross-sectional and intervention studies are shown in Table 1 and Table 2, respectively, and studies using the same dataset were merged into one column and we reported the overall results. All the included studies involved healthy adult participants, with 10 studies involving elderly participants. There were nine cross-sectional studies which compared mind–body experts to controls. Of these, five studies focused on yoga experts and four focused on TCC experts who had regularly practiced for at least three or more years. The remaining six were intervention studies with different durations of the interventions (range 6–24 weeks) and frequency (range 1–5 sessions per week). Of these, four were randomized controlled trials (RCTs), one was a controlled trial, and one was a “before-and-after” study with no control group. In addition, three studies applied TCC, two studies applied yoga, one applied BDJ, and one applied a multimodal intervention comprising cognitive training, TCC exercise, and group counseling. These intervention studies examined the brain health outcomes at baseline and the end of the intervention.

### 3.2. Quality Assessment

The quality assessment results for all the cross-sectional and intervention studies are shown in Table 3 and Table 4, respectively, and studies using the same dataset were merged into one column and the overall results are reported. Most of the included cross-sectional studies were characterized as having low risk of bias (*n* = 8), with only one being categorized as having moderate risk of bias. Among these studies, the lack of a clear description of the measurement approach of the exercise conditions was a common problem (*n* = 7). The quality of the intervention studies most often ranged between good and excellent (*n* = 5), with only one being of fair quality. The most frequent missing point was the description and application of instructor or assessor blinding (*n* = 6).

### 3.3. Changes in Brain Regions

#### 3.3.1. Prefrontal Cortex

Structural and functional differences in sub-regions in the prefrontal cortex (PFC) were reported by various studies. Most prominently, the dorsolateral PFC (dlPFC) and the medial PFC (mPFC) were affected by mind–body exercises in the included studies. Structural changes in the PFC were reported in four cross-sectional studies. Compared with novices, TCC experts [36] had a greater cortical thickness (CT) in the right middle frontal sulcus (part of the dlPFC) and yoga experts [41] showed greater CT in the left prefrontal lobe cluster that included part of the middle and superior frontal gyri (MFG and SFG). Greater gray matter volume (GMV) was reported in the orbitofrontal cortex (OFC), right MFG, and the mPFC in yoga experts compared with controls, which were positively correlated with fewer cognitive failures [32,38].

Changes in task-induced brain activation were reported in three cross-sectional studies and one RCT study. For the cross-sectional studies, less activation was reported in the dlPFC during the Sternberg working memory task [43] and in the left SFG during the Affective Stroop task [33] in yoga experts relative to controls. Besides, less dlPFC activation was reported regarding the N-back task in the TCC experts compared with the water aerobics practitioners [45]. However, in the RCT study, there was an increased trend of left SFG activation following the TCC intervention during the modified task-switching fMRI paradigm. Intriguingly, the TCC group with greater PFC activation in the switch condition reported better cognitive function [57].

Two cross-sectional and two RCT studies reported changes in spontaneous brain neural activity. In the cross-sectional studies, compared with novices, the TCC experts had less regional homogeneity (ReHo) in the dlPFC and lower voxel-mirrored homotopic connectivity (VMHC) as well as greater fractional amplitude of low-frequency fluctuations (ALFF or fALFF) in the MFG [34,37,46]. Moreover, two RCT studies reported increased ALFF in the dlPFC after a 6-week multimodal intervention including TCC and after a 12-week TCC intervention [50,53]. Tao et al. also reported increased fALFF in the mPFC after a 12-week BDJ intervention [53].

#### 3.3.2. Hippocampus/Medial Temporal Lobe

Two cross-sectional and three intervention studies reported structure changes in the hippocampus and medial temporal lobe (MTL) following mind–body exercise. Compared with novices, greater hippocampal GMV was observed in experienced yoga experts [32,43]. Both the 12-week TCC and BDJ interventions increased the GMV in the hippocampus/MTL, and the increased GMV was positively associated with improved memory abilities [55]. Increased hippocampal GMV (compared with controls) and GMD (compared with active sport and passive groups) were reported in two yoga intervention groups, respectively [48,59].

#### 3.3.3. Lateral Temporal Lobe

In addition to the hippocampus/MTL, structural changes in the lateral temporal lobe, majorly constituting the superior temporal gyrus (STG) and the middle temporal gyrus (MTG), were reported in three cross-sectional and two RCT studies. Here, TCC experts displayed greater CT in the left STG [36] and yoga experts exhibited greater GMV in the left STG [32]. In addition, 8-week TCC intervention increased GMV in the left STG and the right MTG relative to controls and the aerobic exercise group, respectively [58]. Moreover, significantly changed ReHo in the left STG and MTG was reported following 6-week multimodal intervention including TCC, which both correlated with better cognitive function [51].

#### 3.3.4. Insula

One RCT and three cross-sectional studies reported on structural changes of the insula, including great CT and GMV. Increased CT in insula was observed in TCC experts [36] and yoga experts [38] compared with controls. Increased GMV in insula was also reported in yoga experts [32] and TCC and BDJ intervention groups [55]. In addition, the reported structural changes in the insula uniquely correlated with pain tolerance in yoga experts [38].

#### 3.3.5. Cingulate Cortex

Only one RCT and two cross-sectional studies reported structural or functional changes in the cingulate cortex. Compared with controls, yoga experts [38] exhibited both greater CT and GMV in the cingulate cortex, and TCC experts [34] had less ReHo in the left anterior cingulate cortex (ACC). Meanwhile, 8-week TCC intervention increased GMV in the left precuneus/posterior cingulate cortex (PCC) [58].

#### 3.3.6. Other Regions

Changes in other brain regions were reported in the included studies, including the occipital cortex [32,45,58], precentral and postcentral gyri [32,34,36], cerebellum [32,49,50], putamen/caudate [40,55], and the amygdala [55], though fewer results were reported. Several studies also found that changed brain regions were positively correlated with better cognitive function [32,34,55].

### 3.4. Changes in Brain Functional Connectivity and Network

Two cross-sectional and three RCT studies reported differences or changes in brain functional connectivity. Compared to controls, a higher resting-state functional connectivity (rsFC) between the mPFC and right angular gyrus was reported in the yoga experts [42], whereas a weaker rsFC between the dlPFC and MFG was observed in the TCC experts [44]. For the RCT studies, significantly greater rsFC was observed between the left MFG and superior parietal lobule (SPL) after 8-week TCC intervention [58], as well as between the mPFC and MTL/hippocampus after 6-week multimodal intervention including TCC [49]. Another RCT study with different a priori regions of interest (ROIs) (dlPFC, hippocampus, and mPFC/PCC) using a seed-based rsFC analysis reported decreased rsFC between the dlPFC and left SFG and ACC after a 12-week TCC intervention, as well as decreased rsFC between the dlPFC and left putamen and insula compared to control groups [52]. At the same time, both the TCC and BDJ interventions increased the rsFC between the hippocampus and mPFC, and there was no significant difference between the two groups [54]. Furthermore, the BDJ intervention decreased the rsFC between the mPFC and OFC/putamen. Whereas, the TCC intervention increased the rsFC between the PCC and right putamen/caudate, and the baseline rsFC between the mPFC and OFC was negatively correlated with memory function [56].

Besides the aforementioned FC results, a few brain networks were reported. TCC experts had less fALFF in the bilateral frontoparietal network (main part of the cognitive control network, CCN) and the default mode network (DMN) [35]. Moreover, one cross-sectional study used a group independent component analysis (gICA) to examine the different correlations of intrinsic connectivity network and found that there were significant differences in the DMN and sensory-motor network (SMN) of rsFC between the TCC and walking groups [47]. Froeliger et al. also reported that yoga experts exhibited greater rsFC within the dorsal attention network (DAN) [31].

## 4. Discussion

Herein, we systematically reviewed evidence of the effect of mind–body exercise on the structure and function of the brain to further understand the possible neural mechanisms underlying the health benefits. We found that both long-term and relatively short-term mind–body exercises induced structural or functional changes, mainly in the PFC, hippocampus/MTL, lateral temporal lobe, insula, and the cingulate cortex and within the CCN and DMN.

### 4.1. Brain Regions

#### 4.1.1. Prefrontal Cortex

The PFC is one of the later-mature brain regions and plays a critical role in a series of advanced cognitive activities [60]. There were inconsistent results regarding the structure of and task-related activation in the PFC between the included cross-sectional and intervention studies. First, four included cross-sectional studies reported structural changes in the PFC, including greater GMV and CT [32,36,38,41]. This implies that long-term mind–body exercise (lasting for at least 3 years of regular practice) can cause positive plasticity in the PFC structure, hence improving the cognitive performance of the individual. However, there were no changes in the PFC structure in the included intervention studies. We speculated that this could be because the relatively short mind–body intervention duration was insufficient for significant changes in the PFC region. Besides, three included cross-sectional studies reported less task-related activation in the PFC, whereas one RCT study reported more [33,43,45,57]. This might be attributed to two different theoretical frameworks that explain the neural plasticity induced by exercise, i.e., reduced neural activity could reflect improved neural processing efficiency, and increased neural activity could indicate more specialized and enhanced neural processing ability [57]. Based on this, we speculate that the relatively short-term mind–body intervention (e.g., the 12-week TCC) could have enhanced the neural processing ability of the PFC, resulting in the greater task-related activation. However, with the continuous increase in regular exercise (e.g., more than 3 years), the neural processing efficiency of the PFC could become more and more efficient, resulting in a subsequent decreased task-related activation in the PFC. Taken together, RCT studies of further prolonged duration should be conducted to clarify the effect of intervention duration on the structure and functional activation of the PFC region.

The findings of the spontaneous neural activity studies indicated that mind–body exercise could decrease the regional homogeneity and increase the functional specialization and intrinsic activity intensity of the PFC, improving the cognitive function [34,37,46,50,53]. Nevertheless, the sub-regions in the PFC affected by different types of mind–body exercises might not be the same. Particularly, TCC affected the spontaneous neural activity of the dlPFC, whereas BDJ affected the mPFC. This difference could be attributed to the different characteristics related to TCC and BDJ. Compared with TCC, which is much more complex and requires moving the trunk and four limbs by spatial navigation toward oneself [36], BDJ is much simpler and only involves eight simple fixed movements of arms with almost no movement of legs [61].

#### 4.1.2. Hippocampus/Medial Temporal Lobe

Both cross-sectional and intervention studies reported the effect of mind–body exercise on structural changes in the hippocampus/MTL [32,43,48,55,59]. Notably, the positive changes in the hippocampus/MTL structure were related to the improvement in the cognitive function, particularly the memory function [32,55]. The hippocampus plays a crucial role in learning and memory processing [62], and the MTL, consisting of the hippocampus, adjacent parahippocampal, and entorhinal cortex, is also essential for memory processing [63], which shows that different types of mind–body exercises can improve the memory of an individual by changing the structure of the hippocampus/MTL. The effects of mind–body exercises on the hippocampus are similar to the findings of physical activity [64] and mindfulness meditation [65], suggesting that physical activity alone or meditation alone, as well as the combination of both, improves the brain structure related to memory. In addition, the intervention study [59] conducted by Garner et al. indicated that yoga experts have a significantly increased GMD in the hippocampus compared with a sport group (stretching and strengthening training), implying that mind–body exercises are more effective in improving the hippocampal structure than physical exercises alone. However, because of the methodological defects of the study (participants were not randomly allocated), the baseline hippocampal GMD in the yoga group was significantly lower than in the sport group. Therefore, the efficacy of mind–body exercise on hippocampal structure should be further explored using methodologically sound RCT studies.

#### 4.1.3. Lateral Temporal Lobe

Several included studies reported structural and functional changes in the STG and MTG in the mind–body exercise groups compared to controls or individuals performing aerobic exercises [32,36,51,58]. The STG is thought to be sensitive to emotional information and plays an important role in a goal-directed behavior [66]. The MTG has extensive connectivity with the frontal parietal regions and is associated with semantic and memory processing [67]. A recent meta-analysis of RCT studies showed that physical activity significantly increases the structure of the lateral temporal lobe [68]. Our results revealed that the mind–body exercises equally or, in some cases, more effectively induced brain plasticity in the lateral temporal lobe, leading to more functional improvements than other forms of exercise, such as aerobics.

#### 4.1.4. Insula and Cingulate Cortex

Mind–body exercises mainly induced structural changes in the insula and cingulate cortex [32,36,38,42,55]. The insula is believed to be related to interoceptive awareness [69] and the integration of sensory information to produce emotional experiences [70], as well as pain processing [71]. Meanwhile, the extensive connection between the cingulate cortex (particularly the ACC) and the PFC and insula is linked to emotion regulation and attention control [72,73]. Young et al. showed that changes in ACC activity reflect the mental processing of non-judgmental acceptance, one of the core components of meditation training [60]. Besides, the ACC is also correlated with pain procession [74]. Villemure et al. indicated that the insula GMV is associated with pain tolerance, with yogis using mental strategies based on relaxation, focusing on the pain, and non-judgmental acceptance to tolerate pain [38]. Combined with the authors’ interpretations, we believe that those changes could be attributed to the inclusion of meditation training during the mind–body exercise, including present moment awareness and non-judgmental acceptance. Moreover, the repeated and prolonged use of these strategies causes positive plasticity in the related brain regions such as the insula and ACC. This further enhances the level of meditation of the individual and corresponding tolerance to noxious stimuli such as pain. Even so, relatively few included studies reported changes in these two regions; hence, more research is required to elucidate this issue.

### 4.2. Brain Networks

Mind–body exercise not only affects a specific brain region, but also the interconnection among different brain regions and various brain networks. This allows more meticulous access to the underlying neural mechanisms. The rsFC results of the included studies mainly reported changes between the PFC (two major sub-regions: dlPFC and mPFC) and other related regions. There were four a priori ROIs in the included rsFC analysis studies: the dlPFC, mPFC, PCC/precuneus, and the hippocampus/MTL. The dlPFC is a vital region of the CCN which plays an essential role in cognitive control processes [75]. The mPFC, PCC/precuneus, and MTL/hippocampus constitute the main nodes of the DMN which are functionally relevant to internal mental explorations and memory function [76,77].

#### 4.2.1. Cognitive Control Network

We found weaker rsFC within the CCN of older TCC experts [44], as well as older TCC and BDJ intervention groups [52]. Similarly, a weaker fALFF in the CCN was reported in older TCC experts [35], and the decreased rsFC and fALFF within the CCN were correlated with better cognitive control and emotion regulation [35,44,52]. In contrast, there was an increased FC within the CCN after TCC intervention in young college students [58]. The conflicting results may be attributed to the different ages of the participants, to some extent. In the first three studies, all of the participants were elderly. Research shows that old people frequently over-recruit frontal neural resources to compensate for the disruption of the CCN and to overcome cognitive control deficits [78]. However, this hyperactivation usually represents a dysfunctional condition and cannot prevent a decline in cognitive function [79]. The decreased rsFC or fALFF in the CCN of the old mind–body exercise experts might suggest an increased efficiency of the cognitive control system and eliminate the need for compensatory hyperactivation of the network affected by mind–body exercise. However, Cui et al. assessed young college students whose executive function is still undergoing development, and the increased FC within the CCN after TCC exercise reflects the specialized and enhanced cognitive control. Furthermore, some studies showed that the beneficial effect of physical exercise on the CCN and cognitive function has been widely reported [80]. Similarly, mindfulness meditation enhances self-regulation through three components, i.e., attention control, emotion regulation, and self-awareness [81]. It is reasonable to imply that mind–body exercise, with both the physical activity and meditation components, induces more effective functional changes in the CCN to improve self-regulation in both young and older adults.

#### 4.2.2. Default Mode Network

Consistently increased rsFC within the DMN was reported in yoga experts [42], a 6-week multimodal intervention including TCC [49], and 12-week TCC and BDJ interventions [54]. Besides, the increased rsFC within the DMN was associated with better general cognition and memory function [49,54]. These findings suggest that mind–body exercises, including TCC and BDJ, enhance cognitive function through the same neural mechanisms by increasing intrinsic connectivity between the mPFC and hippocampus/MTL within the DMN. On the contrary, the study of Liu et al. [56] selected different ROIs (mPFC and PCC) and different methods for multiple comparison correction based on the same dataset, and reported different results, including changes in the direction and related regions. This implies that both TCC and BDJ could improve memory function through other different neural circuits, e.g., decreased rsFC between the mPFC and OFC. With the different results taken into account, we speculate that the DMN is a complex brain network encompassing different vital brain regions (e.g., mPFC and PCC), and mind–body exercise could modulate various features of the DMN, through its multifaceted nature that combines movement, breathing, and attention. Besides, the characteristics among different types of mind–body exercise, such as TCC and BDJ, are not exactly the same, which could affect the DMN through different neural mechanisms. Furthermore, as mentioned above, the two articles [54,56] used different methods for multiple comparison correction based on the same dataset. The former applied the family-wise error (FWE) correction method, whereas the latter utilized the false discovery rate (FDR), which might have different influences on the results.

### 4.3. Limitations

Our study faced several limitations. Firstly, because of the wide variety of outcome measures (structure, neural activity, and functional brain connectivity), different study designs, and some studies reporting ROI analyses, there was no quantitative meta-analysis such as activation likelihood estimation (ALE) applied. Secondly, given the relatively few related studies, our systematic review included cross-sectional studies. Nevertheless, due to the inherent defects, the cross-sectional studies could not certainly attribute the group differences in brain structure and function to mind–body exercise. Although the age, gender, and years of education between groups were matched in the included studies, other confounders could still influence the results. For instance, people with a specific brain activation pattern may have the tendency to engage in mind–body exercises. Besides, there is no long-term follow-up study yet, which would be very meaningful to clarify the long-term health effects affected by mind–body exercise. Thirdly, there was a selection and reporting bias due to the focus on a priori defined ROIs rather than whole-brain analyses, particularly in the rsFC analysis studies. Therefore, it is impossible to fully uncover the impact of mind–body exercises on brain plasticity. Fourthly, we still cannot ascertain the role of the physical and mental components of mind–body exercise on brain structural and functional changes due to the lack of direct comparisons between these components. Finally, because of the few relevant studies, we did not include studies of clinical populations with cognitive issues, mood disorders, etc. As an important non-pharmacological therapy, mind–body exercises have shown promising effects on older adults with cognitive impairments [82] and persons with depression [83]. Therefore, to clarify the influence of mind–body exercise on the brain plasticity of clinical populations is an important direction for future studies.

## 5. Conclusions

In the present study, 15 studies which employed MRI to investigate the effects of mind–body exercise on brain plasticity were included. Our synthesis of results revealed that mind–body exercises induced changes in the structure, neural activity, or functional connectivity in various regions of the brain, primarily the PFC, hippocampus/MTL, lateral temporal lobe, insula, and the cingulate cortex, as well as brain networks, including the CCN and the DMN. These changes were associated with health benefits for healthy adults. However, due to the heterogeneity in the study designs, varied age of participants, exercise types, outcome measures, and a priori ROIs, there were some inconsistent results among the included studies. Therefore, findings of this study should be interpreted with caution. RCTs with rigorous designs and similar measured outcomes, as well as whole-brain analyses, should be conducted to unravel the precise underlying neural mechanisms of mind–body exercise.

## Figures and Tables

**Figure 1 brainsci-11-00205-f001:**
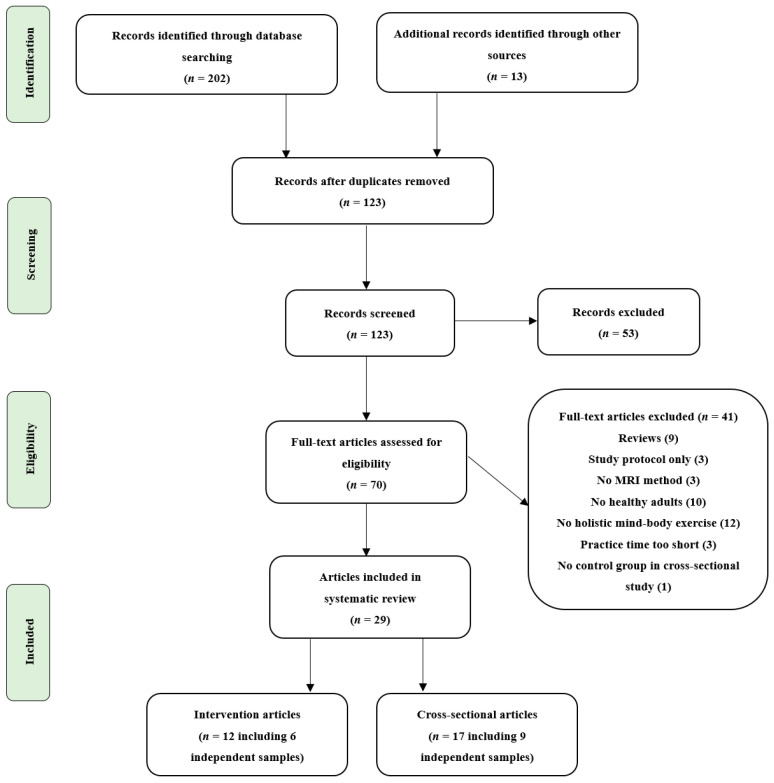
Flow chart of the study selection process for the systematic review.

**Table 1 brainsci-11-00205-t001:** Summary of included cross-sectional studies.

Study	Sample Size (Female)	Group Age in Years (SD)	Group Description	Main Outcome Measures	Primary MRI Results	Association with Behavior Results
Froeliger, 2012a, 2012b, 2012c [31,32,33]	Yoga: 7 (6) CON: 7 (6) *n* = 14 (healthy adults)	Yoga: 36.4 (11.9) CON: 35.5 (7.1)	Yoga participants reported maintaining an active and ongoing modern Hatha yoga practice (>45-min per day, 3–4 times per week, >3 (9.4 ± 2.4) years, and had 5.6 ± 4.2 years of meditation. The age-, sex-, years-of-education-matched CG reported no current or past dedicated meditation or yoga practice.	Cognitive Failures Questionnaire, Affective Stroop task with fMRI, sMRI, R-fMRI	Compared to CG, the yoga group exhibited greater GMV in frontal (bilateral orbital-frontal, right middle frontal, and left precentral gyri), limbic (hippocampus, insula), temporal (left STG), occipital (right lingual gyrus), and cerebellar; had less Stroop BOLD response in left SFG; as well as exhibited greater rsFC within the DAN (ROI).	GMV in regions identified significant group differences were positively correlated with the duration of yoga practice and fewer cognitive failures.
Wei, 2013, 2014, 2017 [34,35,36], Chen, 2020 [37]	TCC: 22 (15) CON: 18 (10) *n* = 40 (healthy old adults)	TCC: 52.36 (6.88) CON: 54.83 (6.77)	TCC group had 14 ± 8 years of TCC experience and 11 ± 3 h per week (TCC styles mainly included Yang, Wu, Sun, and modified Chen). CG had no physical exercise, yoga, or meditation experience. Two groups matched for sex, age, race, and years of education.	ANT, sMRI, R-fMRI	Compared to CG, the TCC group showed significantly thicker cortex in precentral gyrus, insula sulcus, middle frontal sulcus (part of dlPFC) in the right hemisphere and STG, medial occipito-temporal and lingual sulcus in the left hemisphere; had greater ReHo in the right PosCG and less ReHo in left ACC and the right dlPFC; showed lower MFG VMHC; and revealed significantly decreased fALFF in the bilateral frontoparietal network, DMN, and dorsal prefrontal-angular gyri network.	Thicker cortex in left medial occipito-temporal and lingual sulcus was associated with greater intensity of TCC practice, and the reaction time of ANT was positively correlated with cortical thickness of left STG. Increased ReHo in PosCG was correlated with TCC experience, decreases in ReHo in left ACC and increases in ReHo in right PosCG both predicted performance gains on ANT. For TCC practitioners, the longer they practice, the lower their VMHC is in precentral and precuneus. Significant association between TCC practice experience and fALFF in the DMN, as well as an association between cognitive control performance and fALFF of the frontoparietal network.
Villemure, 2014, 2015 [38,39]	Yoga: 14 (9) CON: 14 (9) *n* = 28 (healthy adults)	Yoga: 37.0 (6.6) CON: 36.7 (7.3)	Yoga group had 9.6 ± 2.8 (range 6–16) years of regular yoga practice and 8.6 ± 4.1 h per week (included different yoga styles that integrated postures, breath, and meditation). CG was matched to yogis for sex, age, BMI, education, and exercise level other than yoga postures.	Cold pain tolerance task, sMRI	Compared to CG, the GMV in yogis increased in right cingulate, left dorsal mPFC and insula, and the CT in yogis increased in right insula, right cingulate cortex.	Years of yoga experience correlated with GMV in the insula, frontal operculum and OFC in the left hemisphere, and the number of hours of weekly practice correlated with GMV in SPL, PCC/precuneus, and hippocampus. Combination of postures and meditation contributed the most to the size of the hippocampus, precuneus/PCC, and S1/SPL, and combination of meditation and breathing exercises contributed the most to V1 volume. The insular GMV positively correlated with yoga experience and uniquely correlated with pain tolerance.
Gard, 2015 [40]	Yoga: 16 (69%)Meditators: 16 (63%) CON: 15 (60%) *n* = 47 (healthy adults)	Yoga: 49.38 (7.79) Meditators: 54.06 (8.15) CON: 52.93 (9.84)	Yoga practitioners were primarily trained in Kripalu yoga and had an average of 13,534 ± 9950 h of yoga experience. Meditators were primarily trained in Vipassana meditation and had a 7458 ± 5734 h of meditation experience. CON had less than 4 yoga or meditation classes in the last year and less than 10 classes in their lifetime.	R-fMRI	Compared to CG, yoga practitioners and meditators had greater widespread rsFC (increased degree centrality) of the caudate.	Not reported
Afonso, 2017 [41], Santaella, 2019 [42]	Yoga: 21 (21) CON: 21 (21) *n* = 42 (healthy old women)	Yoga: 66.2 (0.98) CON: 67.9 (1.004)	Yoga group practiced at least twice a week for a minimum of 8 years (yoga style was Hatha). CG was naive to yoga, meditation, or any mind–body intervention and were matched to the yoga group in age, years of formal education, and level of physical activity.	sMRI, R-fMRI	Compared to CG, yoga practitioners showed significantly greater CT in a left prefrontal lobe cluster, which included portions of the lateral MFG and anterior and dorsal SFG; and had a higher rsFC between mPFC (ROI) and the right AGr.	Not reported
Gothe, 2018 [43]	Yoga: 13 (12) CON: 13 (12) *n* = 26 (healthy adults)	Yoga: 35.77 (15.43) CON: 35.69 (14.57)	Yoga group had 9.31 ± 6.25 (range 5–24) years of yoga experience (yoga style was mainly Hatha). CG had no current or past experience with yoga or any other type of mind–body practice. No group differences in demographic measures of income and education and in estimated VO_2_max or physical activity levels.	sMRI, Sternberg Working Memory Task with fMRI	Compared to CG, yoga practitioners had greater GMV in the left hippocampus and revealed less activation in the dlPFC during the encoding phase of the Sternberg task.	Not reported
Liu, 2018 [44]	TCC: 26 (18) CON: 25 (16) *n* = 51 (healthy old adults)	TCC: 65.19 (2.30) CON: 63.92 (2.87)	TCC group had engaged in TCC for an average of 10.44 ± 5.48 years (TCC style was not reported). CG was active in other types of physical exercise without a meditation component, such as jogging and square dancing.Two groups matched for age, gender, years of education, and physical exercise per day.	Five-Facet Mindfulness Questionnaire, Sequential decision task, R-fMRI	Compared to CG, the TCC group showed a weaker rsFC between the dlPFC (ROI) and MFG.	FC between the dlPFC and the MFG in the TCC group fully mediated the impact of non-judgment of inner experience on their emotion regulation ability.
Port, 2018 [45]	TCC: 8 (5) WA: 8 (5) *n* = 16 (healthy old adults)	TCC: 66.4 (7.0) WA: 66.4 (4.9)	TCC (TCC style was not reported) or WA group matched by age, gender, and years of education, had at least 3 years experience and two times a week of practice in TCC or WA.	Stroop Word Color Task and N-back Task with fMRI	Compared to WA group, the TCC group had less brain activation in the right lateral occipital cortex during the Stroop Word Color Task and presented less brain activation in the right frontal pole and SFG during the N-back task.	Not reported
Mei, 2019 [46], Yue, 2020 [47]	TCC: 20 (20) WG: 22 (22) *n* = 42 (healthy old women)	TCC: 62.9 (2.38) WG: 63.27 (3.58)	The TCC group practiced Yang-style TCC 4 ± 1 times weekly for about 1.5 h each time for more than 6 years. WG mainly exercised by walking, not less than 5 times a week, with each time being no less than 1.5 h for more than 6 years. Two groups matched for age, handedness, and years of education.	2-back task, R-fMRI	Compared to WG, The TCC group showed larger fALFF in left MFG; and had significant differences in DMN, SMN, and VN of R-fMRI.	The working memory was correlated with the fALFF of the left MFG in TCC group.

Notes: Studies using the same dataset were merged into one column and the overall results are reported. Abbreviations: ACC = anterior cingulate cortex; Agr = angular gyrus; ANT = attention network test; BMI = body mass index; BOLD = blood oxygenation level-dependent; CG = control group; CON = control; CT = cortical thickness; DAN = dorsal attention network; dlPFC = dorsolateral prefrontal cortex; DMN = default model network; fALFF = fractional Amplitude of Low Frequency Fluctuations; fMRI = functional magnetic resonance imaging; GMV = gray matter volume; MFG = middle frontal gyrus; mPFC = medial prefrontal cortex; OFC = orbitofrontal cortex; PCC = posterior cingulate cortex; PosCG = post-central gyrus; ReHo = regional homogeneity; R-fMRI = resting-state functional magnetic resonance imaging; ROI = region of interest; rsFC = resting-state functional connectivity; S1 = primary somatosensory cortex; SFG = superior frontal gyrus; SMN = sensory-motor network; sMRI = structural magnetic resonance imaging; SPL = superior parietal lobule; STG = superior temporal gyrus; TCC = Tai Chi Chuan; V1 = primary visual cortex; VN = visual network; VMHC = voxel-mirrored homotopic connectivity; WA = water aerobics; WG = walking group.

**Table 2 brainsci-11-00205-t002:** Summary of included intervention studies.

Study	Study Design	Sample Size (Female)	Group Age in Years (SD)	Intervention Frequency and Duration	Main Outcome Measures	Primary MRI Results	Association with Behavior Results
Hariprasad, 2013 [48]	Intervention without control	Yoga: 7 (3) *n* = 7 (healthy old adults)	Age range: 69–81	Yoga: 60 min/day, 5 days/week (yoga style was mainly Hatha), lasted for 3 months + home practice for 3 months.	sMRI	Increased GMV in bilateral hippocampus (ROI) following yoga intervention.	Not reported
Li, 2014 [49], Yin, 2014 [50], Zheng, 2015 [51]	RCT	IG: 17 (8) CG: 17 (6) *n* = 34 (healthy old adults)	IG: 68.59 (5.65) CG: 71.65 (4.00)	IG: Cognitive intervention (MT and EFT): 60-min/session, 3 sessions/week, lasted for 6 weeks; TCC: 60-min/session, 3 sessions/week lasted for 6 weeks (Yang-style 24-form TCC); Group counseling: 90-min/session, 1 session/week, lasted for 6 weeks. CG: two 120-min health-related lectures on health and aging.	Cognitive function: PALT, digit span, TMT, Stroop Test, CFT Social support: SSRS Subjective well-being: SWLS, IWB, R-fMRI	For IG, after intervention, the ALFF was increased in the right MFG, left SFG, left ACL; the ReHo was increased in the left STG, left PCL, and decreased in the left MTG; and the mPFC-MTL (ROIs) FC increased dramatically.	The ALFF changes in the right MFG were positively correlated with changes in TMT and SWLS, and the baseline ALFF in the right MFG was negatively correlated with changes in TMT and SWLS. For IG, the ReHo changes in the left STG were positively correlated with changes in CFT, and changes in the right MTG were negatively correlated with changes in total PALT. For IG, changes for the mPFC-MTL FC were positively correlated with changes in CFT.
Tao, 2016, 2017a, 2017b, 2017c [52,53,54,55], Liu 2019 [56]	RCT	TCC: 21 (13) BDJ: 16 (10) CON: 25 (19) *n* = 62 (healthy old adults)	TCC: 62.38 (4.55) BDJ: 62.18 (3.79) CON: 59.76 (4.83)	TCC: 60 min/session, 5 sessions/week, lasted for 12 weeks (Yang-style 24-form), each session included warm-up and review of TCC principles (10 min), TCC exercises (30 min), breathing techniques (10 min), and relaxation (10 min). BDJ: 60 min/d, 5d/week, lasted for 12 weeks, the whole set of BDJ contains 10 postures, including the starting and ending postures; the time schedule was same as the TCC. CON: received basic health education at the beginning and kept original physical activity.	WMS-CR, sMRI, R-fMRI	Compared to CON, after 12nweeks of intervention, results showed both TCC and BDJ could significantly increase GMV in the insula, MTL/hippocampus, amygdala, and putamen, and no significant differences were observed between the two groups. TCC increased fALFF in the dlPFC and BDJ increased fALFF in the mPFC in the slow-5 and low-frequency bands; both TCC and BDJ (at a lower threshold) significantly increased the rsFC between the bilateral hippocampus (ROI) and mPFC (ROI) and no significant difference between the two groups was observed; TCC showed a significant decrease in rsFC between the dlPFC (ROI) and the left SFG, ACC, and insula/putamen (at a lower threshold); the BDJ showed a significant decrease in rsFC between the dlPFC(ROI) and the left putamen and insula and ACC (at a lower threshold), and there was no significant difference between the two groups. TCC increased rsFC between the PCC (ROI) and the right putamen/caudate, while BDJ decreased rsFC between the mPFC (ROI) and right orbital prefrontal gyrus/putamen and left ACC; compared to BDJ, TCC increased rsFC between the mPFC (ROI) and right putamen/caudate.	MQ and visual reproduction subscores were both associated with GMV increases in the putamen and hippocampus. Increased fALFF in both TCC and BDJ groups was positively associated with memory function improvement. The increases in rsFC between the bilateral hippocampus and mPFC were significantly associated with memory function improvement across all subjects. Mental control improvement was negatively associated with rsFC dlPFC-putamen changes across all subjects. Baseline visual reproduction subscore was negatively correlated with rsFC between the mPFC and right orbital prefrontal gyrus.
Wu, 2018 [57]	RCT	TCC: 16 (13) CON: 15 (15) *n* = 31 (healthy old adults)	TCC: 64.9 (2.8) CON: 64.9 (3.2)	TCC: 60 min/session, 3 sessions/week, lasted for 12 weeks (24-form Yang-style TCC), each session consisting of warm-up (10 min), new TCC form learning (10 min), continuous sequential practice of learned forms (30 min), and cool-down (10 min). CON: maintained original daily routines and physical activity, only received one telephone consultation biweekly.	IED test, task-switching behavioral measures with fMRI	TCC group had an increased trend in the left SFG (ROI: PFC) activation (contrast: switch > non-switch) after intervention.	TCC group with greater PFC activation increases in the switch condition presented greater reductions in task-switching errors from pre- to post-intervention.
Cui, 2019 [58]	RCT	TCC: 12 (10) AE: 12 (10) CON: 12 (10) *n* = 36 (healthy adults)	TCC: 21.83 (2.48) AE: 21.92 (2.28) CON: 21.75 (2.45)	TCC: 60 min/session, 3 sessions/week, lasted for 8 weeks (Bafa Wubu of TCC), each session including warm-up (5 min), continuous sequential practice (50 min), cool-down (5 min). AE: 60 min/session, 3 sessions/week, lasted for 8 weeks, brisk walking. CON: maintained original daily routines and physical activity habits and did not receive any new or additional exercise interventions.	sMRI, R-fMRI	Compared to CG, GMV in the TCC group was significantly increased in the left MOG, left precuneus, left STG, and right MTG, and compared with AE group, TCC group increased the GMV in left MOG, left STG, and right MTG; significant rsFC increases between the left MFG (ROI) and left SPL in the TCC group and no significant rsFC differences were observed in the AE and control groups.	Not reported
Garner, 2019 [59]	NRCT	Yoga: 39 (34) SG: 32 (31) PG: 31 (21) *n* = 102 (healthy adults)	First cohort: 22.7 (2.3) range: 18–29 Second cohort: 22.9 (4.4) range: 18–49	Yoga: 75 min/session, 1 session/week, lasted for 10 weeks (yoga style was Hatha), each session including breathing exercise (10 min), IRT (2 min), loosening exercise (5 min), QRT (3 min), Surya Namascar (10 min), Asanas (25 min), DRT (10 min), and meditation (10 min). SG: 45–60 min/session, 1 session/week, lasted for 10 weeks, anaerobic fitness-oriented stretching and strengthening training program. PG: maintained daily habits.	sMRI	Increased right hippocampal GMD among yoga group	Not reported

Notes: Studies using the same dataset were merged into one column and the overall results are reported. Abbreviations: ACC = anterior cingulate cortex; ACL = anterior cerebellum lobe; AE = aerobic exercise; BDJ = Baduanjin; CG = control group; CON = control; CFT = category fluency test; dlPFC = dorsolateral prefrontal cortex; DRT = Deep Relaxation Technique; EFT = executive function training; (f)ALFF = (fractional) Amplitude of Low Frequency Fluctuations; fMRI = functional magnetic resonance imaging; GMD = gray matter density; GMV = gray matter volume; IED = Intra-/Extra-Dimensional Set Shift; IG = intervention group; IRT = Instant Relaxation Technique; IWB = index of well-being; MFG = middle frontal gyrus; MOG = middle occipital gyrus; mPFC = medial prefrontal cortex; MQ = memory quotient; MT = mnemonic training; MTG = middle temporal gyrus; MTL = medial temporal lobe; NRCT = non-randomized controlled trial; PALT = paired associative learning test; PCC = posterior cingulate cortex; PCL = posterior cerebellum lobe; PFC = prefrontal cortex; PG = passive group; QRT = Quick Relaxation Technique; RCT = randomized controlled trial; ReHo = regional homogeneity; R-fMRI = resting-state functional magnetic resonance imaging; ROI = region of interest; (rs)FC = (resting state) functional connectivity; SFG = superior frontal gyrus; SG = sport group; sMRI = structural magnetic resonance imaging; SPL = superior parietal lobule; SSRS = social support rating scale; STG = superior temporal gyrus; SWLS = subjective well-being; TCC = Tai Chi Chuan; TMT = trail making test; WMS-CR = Wechsler Memory Scale—Chinese revision.

**Table 3 brainsci-11-00205-t003:** Quality assessment of the included cross-sectional studies.

	A	B	C	D	E	F	G	H	OQ
Froeliger, 2012a, 2012b, 2012c [31,32,33]	+	+	?	+	+	+	+	+	Low risk
Wei, 2013, 2014, 2017 [34,35,36] Chen, 2020 [37]	+	+	?	+	+	+	+	+	Low risk
Villemure, 2014, 2015 [38,39]	+	+	+	+	+	+	+	+	Low risk
Gard, 2015 [40]	+	?	?	+	+	+	+	+	Low risk
Afonso, 2017 [41], Santaella, 2019 [42]	+	+	?	+	+	+	+	+	Low risk
Gothe, 2018 [43]	+	+	+	+	+	+	+	+	Low risk
Liu, 2018 [44]	+	+	?	+	+	+	+	+	Low risk
Port, 2018 [45]	+	?	?	+	+	−	+	+	Moderate risk
Mei, 2019 [46], Yue, 2020 [47]	+	+	?	+	+	+	+	+	Low risk

Notes: Studies using the same dataset were merged into one column and the overall results are reported. A—Were the criteria for inclusion in the sample clearly defined? B—Were the study subjects and the setting described in detail? C—Was the exposure measured in a valid and reliable way? D—Were objective, standard criteria used for measurement of the condition? E—Were confounding factors identified? F—Were strategies to deal with confounding factors stated? G—Were the outcomes measured in a valid and reliable way? H—Was appropriate statistical analysis used? OQ, overall quality; +, yes; −, no; ?, unclear; NA, not applicable.

**Table 4 brainsci-11-00205-t004:** Quality assessment of the included intervention studies.

	EC	RA	CA	SAB	SB	IB	AB	DR	ITA	BC	PM	MQ
Hariprasad, 2013 [48]	+	NA	NA	NA	+	NA	NA	+	+	NA	+	Fair
Li, 2014 [49], Yin, 2014 [50], Zheng, 2015 [51]	+	+	+	+	+	?	+	−	+	+	+	Good
Tao, 2016, 2017a, 2017b, 2017c [52,53,54,55], Liu, 2019 [56]	+	+	+	+	+	?	?	−	+	+	+	Good
Wu 2018 [57]	+	+	+	+	+	?	+	+	+	+	+	Excellent
Cui, 2019 [58]	+	+	+	+	+	?	?	+	+	+	+	Good
Garner, 2019 [59]	+	−	−	−	+	+	?	+	+	+	+	Good

Notes: Studies using the same dataset were merged into one column and the overall results are reported. EC, eligibility criteria; RA, random allocation; CA, concealed allocation; SAB, similar at baseline; SB, subject blinded; IB, instructor blinded; AB, assessor blinded; DR, drop-out rate (<15%); ITA, intention-to-treat analysis; BC, between-group comparison; PM, points measures; MQ, methodological quality; +, yes; −, no; ?, unclear; NA, not applicable.

## Data Availability

All data were already included in the main text of the manuscript.

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
