# Peer review of "Effects of Mind–Body Exercise on Brain Structure and Function: A Systematic Review on MRI Studies"

_brainsci, 2021, doi:10.3390/brainsci11020205_

Round 1

Reviewer 1 Report

Abstract:

Please insert the full MRI words before the abbreviation.

Introduction section:

Please, explain what does long-term mind body exercises and relatively short term mind-body exercises consist.

It is a good study to describe the specific regions or networks of the brain affected by exercise, however, the objective of the study: “to elucidate the neural mechanisms that underlie the health benefits of mind-body exercises” has no answer. I suggest rephrasing the objective of the review.

It would be important to add a paragraph explaining more detail about the MRI.

Results section:

Tables 2 and 3 are too large and difficult to read. I suggest breaking it down and organizing your studies with other subsections, such as Changes in Brain Regions, as described below.

Discussion section:

I suggest structuring the discussion in subsections following the order of presentation of the results for a better understanding.

Reviewer 2 Report

Review

Zhang et al. Effects of mind-body exercise on brain structure and function: A systematic review on MRI studies

Overall

In general, the manuscript should be revised in terms of linguistics and grammar to make the writing clearer and more concise – perhaps a native speaker could proofread the manuscript. Search strategy and study selection are sufficiently described and the additional citation search is very good. I think the methodology is thoroughly described but the writing should be improved to convey the content more clearly. Furthermore, I think that the nature of the interventions should be described in more detail as most readers are not aware of their specifics and how meditational/awareness practices are integrated. The discussion needs some restructuring to clearly inform about the results and their implications. Please see the detailed comments for specifics. 

Abstract

Lines 12-13 “To our knowledge, the neural mechanisms behind the improvements in health are still unclear.” Maybe something like “…there is no clear consensus on the neural mechanisms underlying improvements in health/mental health”

Introduction

Line 38 “[…] improving the global cognition […]” better: general cognition or just cognitive function

Lines 49-51 “Researchers have employed non-invasive neuroimaging techniques to identify changes in brain plasticity induced by mind-body exercise. Among these technologies, MRI with good spatial resolution could reveal the changes in the cortical and subcortical region; hence, frequently used in brain science research[18].” These sentences would sound better as in: The effects of mind-body exercise on brain plasticity haven been examined by magnetic resonance imaging (MRI). MRI is a frequently used non-invasive neuroimaging technique with remarkable spatial resolution that allows for investigation of change in cortical as well as subcortical brain regions.

Lines 61-63 “The three available systematic reviews on Tai Chi Chuan[20] and Yoga[17,21], respectively, integrated a few literatures and did not delve into the specific brain regions or networks affected.” Maybe like this: The systematic reviews on Tai Chi Chuan[20] and Yoga[17,21], were based on a small number of studies and did not focus on specific brain regions or networks affected by mind-body exercise.

Methods

Line 103 Why did you limit the duration of mind-body exercise to a minimum of 3 years specifically? Could you add a reference that deals with induced brain plasticity and its time course?

Lines 116-127 The whole section is somewhat unclear and should be edited. To me it does not become clear from the references that you cited on how the cutoff values were determined. Please elaborate a bit more on that.

Results

Line 134 Figure 1 does not represent the PRIMSA checklist but rather a flowchart that is structured based on the study selection recommended by PRISMA

Table 1, 2 Please state in the description of the table that you merged studies using the same cohort into one column and report the overall results.

Lines 183-185 “Structural and functional changes in the prefrontal cortex (PFC), involving different sub-regions, particularly the dorsolateral PFC (dlPFC) and the medial PFC (mPFC) were the most reported regions that were influenced by mind-body exercises in the included studies.” This should be modified to “Structural and functional differences (as you refer to both intervention as well as cross-sectional studies) in sub-regions in the prefrontal cortex (PFC) were reported by various studies. Most prominently the dorsolateral PFC (dlPFC) and the medial PFC (mPFC) were affected by mind-body exercises in the included studies.”

Line 192 “The changes of task-induced brain activation […]” Change to “Changes in […]” as it refers to multiple studies.

Lines 237-239 “Changes in other regions of the brain were reported in the included studies, including the occipital cortex, precentral/postcentral gyrus, cerebellum, and the putamen/caudate, though fewer results were reported.” Please cite the studies that found these results and if any behavioral or psychometric measures were correlated.

Discussion        

Overall, to make the discussion more concise and comprehensible, my suggestion would be to refer to functional aspects of brain regions, e.g. memory, cognitive control, emotion regulation etc. and then report results and implications. That way, you can group findings on functional, structural and connectivity differences in speculated behavioral outcomes. In my opinion, that would make the discussion a lot more comprehensible.

Lines 278-280 “We speculate that this could be because the short duration (6-24 weeks) was not sufficient for the intervention to cause significant changes to the relatively stable PFC region.” Could you please cite a reference that refers to the structural stability of the PFC?

Lines 296-298 “This difference could be attributed to the different characteristics related to TCC and BDJ, in which the movement in BDJ is much simpler than TCC.” It is very interesting that types of mind-body exercise can be contrasted in terms of neural activation but could you please elaborate a bit more on the difference between those exercise types as the average reader might not be familiar with them.

Lines 314-318 “Therefore, whether the mind-body exercise effects on the hippocampus are more effective should be further explored using RCT studies with rigorous designs and randomly allocated subjects to different groups, such as the mind-body exercise group, physical activity group, and the meditation group, by directly comparing the pre- and post- intervention results.” This could be shortened to “Therefore, the efficacy of mind-body exercise on hippocampal structure and function should be further explored using methodologically sound RCT studies”

Lines 363-365 “The decreased rsFC or fALFF in the CCN of the elderly participants might suggest an increased efficiency of the cognitive control system and eliminate the need for compensatory hyperactivation of the network in old mind-body of the experts.” Please rewrite as I assume you mean elderly mind-body practitioners show less compensatory hyperactivation?

Lines 383-387 “Combining these different results within the DMN, we speculate that, firstly, the DMN is a complex brain network encompassing different vital brain regions, including the mPFC and PCC and the mind-body exercise comprises multiple components combining movement, breathing, and attention; hence, modifies the functional activity and connectivity in the DMN via multiple mechanisms to enhance the cognitive function.” Please rewrite to first state that the DMN is a complex network etc. and then go on to say that mind-body exercises modify various features of that network through its multifaceted nature that combines movement, breathing and attention.

Lines 387 – 390 “Besides, because of the similar and different characteristics among the various exercise types, the mind-body exercise could also through the same or different pathways to modulate the complex brain network, such as the DMN.” I am not sure what you mean here, please rewrite.        

Lines 390 -393 “Furthermore, it is worth noting the two articles[46,48] used a different multiple comparison correction method for the same dataset. The former applied the family-wise error (FWE) correction method, while the latter utilized the false discovery rate (FDR); hence, this could have different effects on the results.” Please mention this earlier in line with the reported results in line 381.

Lines 409 – 410 “In the present study, 15 studies, which employed MRI to investigate the effects of mind-body exercise on the brain plasticity changes, were included.” I would omit changes as plasticity inherently suggests change.

Reviewer 3 Report

The study makes a review of different works that make exercise protocols with beneficial effects on mind and body at the same time that they analyze the improvements at brain level with different neuroimaging techniques. The study is interesting and well written. In addition, the methodology used for the selection of the works has been correct. However, before its publication, the work must be improved in some aspects:

-To look for works in which the cognitive benefit of this type of exercise is valued over other functions different from memory. For example, there is literature about the positive effect of exercise on executive functions. On the other hand, it would be interesting to have longitudinal pre- and post-exercise measures, as well as long-term measures to assess how long the benefits of exercise last.

-It would be interesting to also include a small section about the impact of this type of exercise in populations with some type of cognitive impairment or mental disorder. This could be a brief section at the end of the article. 

-Have the authors observed differences based on gender and/or age?

-Reviewing the discussion and conclusion of the study, it is not clear whether the changes seen at the brain level are due to mental or physical exercise or a combination of both, since in all cases changes in brain areas such as the hippocampus are described. This point should be clarified.

Round 2

Reviewer 2 Report

Dear authors,
thank you for the meticulous revision of your manuscript. I am pleased to see that all my suggestions as well as other reviewers comments were implemented. Previously unclear parts of the article are now comprehensible and the language revision makes it a much improved and thoroughly written review.

Reviewer 3 Report

The authors have responded to all my comments and have included them in the manuscript. I therefore accept the article in its revised version.